# Fungal Species and Multi-Mycotoxin Associated with Post-Harvest Sorghum (*Sorghum bicolor* (L.) Moench) Grain in Eastern Ethiopia

**DOI:** 10.3390/toxins14070473

**Published:** 2022-07-11

**Authors:** Abdi Mohammed, Zelalem Bekeko, Mawardi Yusufe, Michael Sulyok, Rudolf Krska

**Affiliations:** 1School of Plant Sciences, College of Agriculture and Environmental Sciences, Haramaya University, Dire Dawa P.O. Box 138, Ethiopia; zelalembekeko@yahoo.com; 2Institute of Technology, Food Sciences and Post-harvest Technology, Haramaya University, Dire Dawa P.O. Box 138, Ethiopia; mawryusuf@yahoo.com; 3Institute of Bioanalytics and Agro-Metabolomics, Department of Agrobiotechnology (IFA-Tulln), University of Natural Resources and Life Sciences, Vienna Konrad Lorenzstr. 20, A-3430 Tulln, Austria; michael.sulyok@boku.ac.at (M.S.); rudolf.krska@boku.ac.at (R.K.); 4Institute for Global Food Security, School of Biological Sciences, Queens University Belfast, University Road, Belfast BT7 1NN, UK

**Keywords:** multi-mycotoxins, sorghum grain, eastern Ethiopia

## Abstract

Sorghum is the main staple food crop in developing countries, including Ethiopia. However, sorghum grain quantity and quality are affected by contaminating fungi both under field and post-harvest stage. The aim of the current study was to assessed fungal species and multi-mycotoxins associated with sorghum grain in post-harvest samples collected from eastern Ethiopia. Fungal genera of *Aspergillus*, *Alternaria*, *Bipolaris*, *Fusarium*, *Mucor*, *Penicillium,* and *Rhizoctonia* were recovered in the infected grain. A liquid chromatography-tandem mass spectrometric (LC-MS/MS) was used for quantification of multiple mycotoxins/fungal metabolites. Overall, 94 metabolites were detected and grouped into eight categories. All metabolites were detected either in one or more samples. Among major mycotoxins and derivatives, deoxynivalenol (137 μg/kg), zearalenone (121 μg/kg), ochratoxin A (115 μg/kg), and fumonisin B_1_ (112 μg/kg) were detected with maximum concentrations, while aflatoxin B_1_ had relatively lower concentrations (23.6 μg/kg). Different emerging mycotoxins were also detected, with tenuazonic acid (1515 μg/kg) occurring at the maximum concentration among *Alternaria* metabolites. Fusaric acid (2786 μg/kg) from *Fusarium* metabolites and kojic acid (4584 μg/kg) were detected with the maximum concentration among *Fusarium* and *Aspergillus* metabolites, respectively. Unspecific metabolites were recognized with neoechinulin A (1996 μg/kg) at the maximum concentration, followed by cyclo (L-Pro-L-Tyr) (574 μg/kg) and cyclo (L-Pro-L-Val) (410 μg/kg). Moreover, metabolites form other fungal genera and bacterial metabolites were also detected at varying levels. Apparently, the study revealed that sorghum grains collected across those districts were significantly contaminated with co-occurrences of several mycotoxins. Farmers should be the main target groups to be trained on the improved management of sorghum production.

## 1. Introduction

Sorghum (*Sorghum bicolour* L. Moench) is a tropical cereal crop cultivated in the warmer climatic areas across the world. Ethiopia is considered as one of the greatest sorghum producers, with 2,668,177.68 tons, making it the third greatest in Africa next to Nigeria (7,337,775.96 tons) and Sudan (3,722,277.78 tons) [1]. Globally, sorghum is among the most important nutritional staple grain crops for more than 500 million people, mostly in poor nations, providing carbohydrates, vitamins, protein, and minerals [2]. In Ethiopia, sorghum is grown on approximately 1,679,277.06 hectares of land, with a total annual yield of 4,517,350.21 tons [3]. About 4.5 million smallholders located in the eastern and north-west parts of the country cultivate sorghum. All of the sorghum produced in the country is used for domestic consumption, and its contribution to food security is significant. According to the Central Statistical Agency (CSA) [4], sorghum is the third most important food cereal crop in Ethiopia, after maize and *teff*, in terms of the total number of growers involved, area coverage, and grain production. It is typically used to make *Injera*, a local bread, boiled porridge or gruel, malted beverages, popped grain, as well as homemade local beverages such as *tela*, *areke*, and *bordie* [5]. Sorghum is the second most important crop for quality *Injera* next to *teff* in Ethiopia. 

Post-harvest handling and inadequate storage conditions are the main contributing factors to food crop losses. In sub-Saharan Africa (SSA) countries, post-harvest loss is the main food crop insecurity problem [6]. In Ethiopia, massive amounts (5–26%) of cereal post-harvest losses have been reported due to poor storage conditions and a weak post-harvest system [7]. For instance, according to [8], 27% of harvested sorghum grain in Ethiopia is lost at the post-harvest stage, owing to several factors. The losses that may occur in this chain includes harvesting, drying in the field and/or on platforms, threshing and winnowing, and transport to store, farm storage, market, and market storage. Given that sorghum grain is an ideal substrate for mold development when poorly dried and stored, this subsequently contributes to post-harvest deterioration of grains. A major concern associated with grain mold is the production of mycotoxins [9].

Sorghum is affected by various fungal species in the field and post-harvest stages, known to produce mycotoxins that pose serious health risks to humans and animals that feed on contaminated grains [10,11]. Mycotoxins are natural secondary metabolites produced by fungi that grow on a variety of agricultural products including cereals, nuts, spices, apples, dried fruits, and coffee beans [12]. The majority of mycotoxins are produced by the genera of *Aspergillus*, *Fusarium*, and *Penicillium* species, a concern for food safety in the developed and developing world [13]. These mycotoxins are associated with human diseases such as cancer and hepatitis [14], and the trichothecenes are labeled as bioterrorism agents [15]. Mycotoxin effects in humans and animals following direct exposures varies in terms of their toxicity, e.g., carcinogenic, endocrine disorders, teratogenic, mutagenic, hemorrhagic, estrogenic, hepatotoxic, nephrotoxic, and immunosuppressive [16]. In Ethiopia, different fungal species such as *A. flavus*, *A. niger*, *A. parasiticus*, *Alternaria*, *Fusarium*, *Penicillium,* and others were isolated from sorghum grain [17,18,19]. The most important classes of mycotoxins include the highly carcinogenic aflatoxins (e.g., AFB_1_), trichothecenes (e.g., deoxynivalenol, DON), fumonisins (e.g., FB_1_), ochratoxin A (OTA), and zearalenone (ZEN) [20,21]. These toxins and several others are addressed by regulatory limits in many countries of the world after thorough risk assessment, taking into account toxicity, occurrence, and consumption data as well as economic and political considerations [22]. Furthermore, emerging mycotoxins become investigated due to the food safety concern and health impacts and advanced technology, used for diagnosing trace compounds in food matrix. Primarily, an emerging mycotoxin was reported in 2008 that deals with the *Fusarium* metabolites such as fusaproliferin (FP), beauvericin (BEA), enniatins (ENNs), and moniliformin (MON) [23]. However, data on the toxicity and occurrence of emerging mycotoxins are limited. Nevertheless, there have been some studies describing their potential implications for food safety [24]. Based on the scientific opinion of European Food Safety Authority (EFSA), some opinion papers about the risk to human and animal health to the presence of regulated, modified, and emerging mycotoxins have been published [25,26,27]. Emerging mycotoxins in food crops, and certainly in sorghum grain, are not exhaustively studied in Ethiopia.

Addressing the issue of mycotoxin contamination and exposure in Africa is still a pressing challenge. Among most African governments, food safety is not regarded as a priority, particularly for domestic populations [28]. For instance, in Ethiopia, there is no standard and regulation strategy for tackling mycotoxin contamination in food crops, including cereal crops. In some sub-Saharan African (SSA) countries, including Burkina Faso, Ethiopia, Mali, and Sudan, mycotoxin contamination in sorghum grain was prevalent, in which Ethiopian samples were 11.0% contaminated with several mycotoxins [29]. Among several mycotoxins detected, prevalence of sterigmatocystin, fumonisins, and aflatoxins were reported in the sorghum samples [29]. Mycotoxins from *Aspergillus* and *Alternaria* species are the mycotoxins of concern in SSA grain sorghum with regard to prevalence, concentration, and health risk exposure. Aflatoxin (344 μg/kg) and fumonisin (2041 µg/kg) in the post-harvest sorghum samples were reported from Ethiopia [18]. In Ethiopia, recently, the economic and health impacts of mycotoxin prevalence in different food crops were extensively reviewed [30]. However, such emerging mycotoxins were not heartened, likely due to limitation of exhaustive published data in the country. Indeed, [17] attempted to report multiple mycotoxins in sorghum and finger millet. However, the present study covered wide geographic areas known by sorghum production. Hence, the aim of the current study the determination of major fungal species and multi-mycotoxins associated with post-harvest sorghum grain in eastern Ethiopia, covering East and West Hararghe zones.

## 2. Result and Discussions

### 2.1. Analysis of Fungal Species in Sorghum Grain Samples

Different fungal species were isolated from the contaminated sorghum samples obtained from all districts: *Aspergillus flavus*, *A. niger*, *Alternaria*, *Bipolaris*, *Fusarium*, *Mucor*, *Penicillium*, and *Rhizoctonia* species. *Bipolaris* species was isolated from Fedis and Miesso districts, while *Mucor* species was isolated only in samples collected from Miesso district. The incidences of those fungal species infecting sorghum grain samples across those districts are presented below (Figure 1).

The maximum incidence (37%) of infection was found from Miesso district samples, followed by Fedis (27%), while samples from Goro Gutu were found to be less contaminated (15%). It could be due to Miesso and Fedis being relatively lowland, given that warm weather creates conducive conditions for fungal development, while Goro Gutu has higher elevations (Table 4). However, as a contamination factor, other associated bio-factors were not discussed in the entire sections. Different fungal species are known to contaminate sorghum grains. The taxonomic diversity of the fungal genera contaminating sorghum grain most commonly encompasses, but is not limited to, *Fusarium*, *Aspergillus*, *Curvularia*, *Colletotrichum,* and *Alternaria* species [31], and likely contribute to pre- and post-harvest grain deteriorations. The various fungal infections of the sorghum grain may become complex and fluctuate throughout sorghum growth, harvest, and storage practices. From Ethiopia, [17] revealed species of *Aspergillus*, *Alternaria*, *Fusarium*, *Penicillium*, *Rhizopus* and *Epicoccum*, isolated from sorghum grain. However, in the current study, apart from those dominant fungal species of *Aspergillus*, *Alternaria*, *Fusarium*, and *Penicillium, Bipolaris* and *Mucor* species were identified in samples from different districts and conform to the finding of [17]. The study also conforms to report of [18], which revealed *A. flavus*, *A. niger,* and *A. parasiticus* isolated in sorghum grain of from post-harvest in Ethiopia. Nevertheless, *A. parasiticus* was not recovered in our study.

Regardless of the fungal species prevalence, *A. niger* and *Fusarium* species were the most dominant, followed by *A. flavus*, while *Rhizoctonia* and *Bipolaris* species occurred less (Figure 2). Likewise, a similar study from Egypt revealed that, among *Aspergillus* species, *A. niger* becomes the predominant species of fungal count infecting sorghum grain, followed by *A. flavus* [32]. The authors also reported that *Fusarium* and *A. parasiticus* species represented less frequency [32]. In contrast to this, *Fusarium* species comprised to the dominant fungi. However, in another study, *A. flavus* was the predominant species infecting sorghum grain, while *A. niger* was less contaminated from Northern Shewa of Ethiopia [18]. Indeed, the current study was targeted the East and West Hararghe zones, potential sorghum-producing areas in the country. Thereby, geographical locations used for sample collections might have affect mold development, owing to the fact that ecological requirements are the main factors for fungal development. In support of this study, Ref. [19] revealed that *Aspergillus* and *Fusarium* species were the dominant fungal invasion of sorghum grain, despite being influenced by the trashing and storage period. Thus, the study affirmed that *Aspergillus* and *Fusarium* species caused adversities to sorghum grain production from eastern Ethiopia, which might subsequently lead to mycotoxin contamination and food safety problems, given that sorghum is the leading staple crop for household consumptions.

### 2.2. Contamination of Mycotoxin Groups in Sorghum Grain

In this study, multi-mycotoxin analysis in sorghum samples were employed, and 94 metabolites including bacterial toxins were detected. All samples were contaminated by either one or more metabolites. In total, the detected toxins comprised eight categories (Table 1). These are: major mycotoxins and derivatives (*n*= 13), *Fusarium* metabolites (*n* = 18), *Aspergillus* metabolites (*n* = 13), *Penicillium* metabolites (*n* = 21), *Alternaria* metabolites (*n* = 6), metabolites from other fungal genera (*n* = 8), Bacterial metabolites (*n* = 4), and unspecific metabolites (*n* = 11). In this case, metabolites produced by *Penicillium* species were most abundant, followed by *Fusarium* metabolites. *Alternaria* metabolites and bacterial metabolites were detected in less frequency. Sorghum grain contaminated with mycotoxin produced by *Alternaria*, *Aspergillus*, *Fusarium,* and *Penicillium* species were abundantly reported [9,29]. Yet in the present study, of those mycotoxin groups, metabolites of *Penicillium* fungi, metabolites from other genera, and unspecified metabolites were (100%) prevalent in sorghum grain across all the districts. Thus, our study affirmed that Ethiopian sorghum is contaminated with those groups of fungal origin and bacterial metabolites, which may lead to serious food safety issues due to its toxicity impacts.

Geographically, Fedis and Goro Gutu districts were from East Hararghe zones, while Doba and Miesso were from West Hararghe Zones. Moreover, irrespective of the sampled sources, those from Fedis and Miesso were 100% contaminated by five toxins out of eight toxin groups, while samples from Doba and Goro Gutu districts were contaminated with four out of eight toxin groups (Table 1). Globally, it is estimated that 60–80% of food crops are contaminated with mycotoxin [33]. So, the current study confirmed that sorghum grain in Ethiopia is highly contaminated with mycotoxins and contribute to these predictions. The lowest prevalence (15%) of major mycotoxins was revealed in the Fedis district. Those variations might have been influenced by responsible fungi-produced metabolites and handling practices. Overall, the study affirmed that various mycotoxins are abundantly contaminating Ethiopian sorghum and need further mitigation strategies to halt the health risk exposures associated with sorghum-based food consumptions, given that sorghum is the staple food crop across targeted areas.

### 2.3. Multi-Mycotoxins Detected in Sorghum Samples

Mycotoxins of different groups and specific metabolites produced by diverse fungal species in sorghum grain samples were presented (Table 2). Over the years, there was a report that stated that approximately 25% of world cereal crops are contaminated by mycotoxin; however, according to the recent investigations, it become raised to 60–80% [33]. Researchers revealed that about 72% of feed samples from different parts of the world collected over several years contained mycotoxins [34]. In the present study, all samples (100%) were contaminated with either one or more of mycotoxins, which could be in agreement with those earlier reports [35].

Among the major mycotoxins and derivative groups, zearalenone was the most prevalent, detected in 24% of samples, with a maximum concentration of 121 μg/kg; however, deoxynivalenol contaminated only 6% with a low prevalence rate and had maximum levels of 137 μg/kg amongst, but below the maximum tolerable level of 750 μg/kg [36]. The levels of zearalenone detected in sorghum grain in South Africa (70.5 μg/kg) [35] and Togo (25 μg/kg) [37] were found to be below the current finding. The study confirmed that, in sorghum, zearalenone contamination is more prevalent than deoxynivalenol, similar to the report of [17] from Ethiopia. However, the maximum concentration of deoxynivalenol soured zearalenone, which was in argument with the previous study [17,38]. Overall, in [38], the maximum level of deoxynivalenol was detected in sorghum samples, compared to [17] and the present study. It was also astonishing that aflatoxin B_1_, B_2_, and G_1_ were only detected in 1.25%, occurring in low levels and prevalence. However, one sample became contaminated with the maximum concentration of 23.6 μg/kg of aflatoxin B_1_; indeed, this specific figure was beyond the tolerable levels of aflatoxin B_1_ in cereal food, according to the EU commissions ranging from 2 to 12 μg/kg [39]. Likewise, Ref. [17] detected aflatoxin B_1_ in sorghum beyond the international standards in some samples. The occurrence of aflatoxins groups in low levels in sorghum grain in the present study is in agreement with different previous reports, from Ethiopia [17,38], Nigeria [40], Cameroon [41], and Togo [37]. Moreover, in harmony with our study, AFG_2_ was not detected in some previous studies [41,42]. Establishment of ochratoxin A levels in cereals and other commodities is used to formulate the regulations for setting the maximum tolerable levels. The European Union has set a limit of 5 µg/kg and 3 µg/kg in cereals and cereal products, respectively [43]. In the present study, ochratoxin A was detected in 5% of samples with maximum levels of 115 µg/kg, which surpassed the regulated levels 23-fold However, Ethiopia has no set limits for ochratoxin A in cereals and related products.

However, in the earlier study demonstrated from eastern Ethiopia, a total aflatoxin level (B_1_, B_2_, G_1_, and G_2_) up to 344 μg/kg and aflatoxin B_1_ up to 33 μg/kg were detected in the post-harvest sorghum samples [18], found more worthwhile than the current study. In the same report, a total fumonisin level ranging from 907 to 2041 μg/kg was detected [18]. Later on, the mycotoxin contamination of sorghum in SSA, notably from Ethiopia, detected the mean and maximum aflatoxin B_1_ (42 and 126 μg/kg), as well as fumonisin B_1_ (105 and 259 μg/kg) and B_2_ (74.4 and 106 μg/kg), respectively [29]. In the current study, the maximum levels of fumonisin B_1_ (112 μg/kg) and B_2_ (30.3 μg/kg) were detected, and this affirmed that both aflatoxins and fumonisins concentrations were found below the earlier reports. Similarly, the levels of fumonisins B_1_ and B_2_ were below the maximum tolerable limits set by EU commission countries, which are 1000 and 4000 μg/kg, respectively [36]. It is demonstrated that sorghum contamination by mycotoxins is associated with the contaminated fungi with the sorghum grain [32]. Furthermore, ochratoxin A as high as 115 μg/kg was detected, a level lower than the earlier report (175 μg/kg) [38].

Among *Fusarium* metabolites, moniliformin had the maximum prevalence (93%) followed by siccanol (88%), while enniatin A1 and gibepyron D, had less prevalence (2.5%, each). Consistently, in [17], moniliformin had the maximum prevalence (97%) of *Fusarium* metabolites origin. Furthermore, from the same groups of *Fusarium* metabolites, the mean and maximum concentration of fusaric acid was 204 and 2790 μg/kg, respectively, greater than what is reported in [17] (130 and 239 μg/kg), and the concentration of siccanol was 240 and 2374 μg/kg, which was reported in Ethiopian sorghum for the first time.

Presently, some *Fusarium* and rare *Aspergillus* metabolites are considered as an “emerging toxin” [44]. Those that occur most frequently are enniatins, beauvericin, apicidin, aurofusarin, culmorin, butenolide, fusaric acid, moniliformin, fusaproliferin, produced by *Fusarium* species, and emodin, 3-Nitropropionic acid, by *Aspergillus* species [45]. The maximum incidences of enniatins and beauvericin were detected in food (63 and 80%), feed (32 and 79%), and unprocessed grains (24 and 46%) collected from 2010 and 2014 in 12 European countries [24], an argument with the current report, in which moniliformin had the maximum incidences (93%). 3-Nitropropionic acid (3-NPA) had 61% prevalence with a maximum concentration of 817 μg/kg. Among the enniatin groups A, A1, B, and B2 are the most frequently reported in food and feeds [46]; concordant with our study, enniatins (A1, B, and B1) were detected in sorghum samples, albeit in low concentrations. Previously, [47] revealed the likely frequent contamination of enniatins in cereals and cereal-based foods and urged further investigation of such mycotoxins across diverse geographical regions with different climatic conditions from all over the world. Thus, the present study affirmed the contamination of enniatins as an emerging mycotoxin in sorghum grain from Ethiopia. Moniliformin is known to be a worldwide natural contaminant in cereals such as rice, wheat, oats, barley, rye, and maize [48], which entirely supported our finding. Moniliformin was firstly reported in 1982 in moldy maize from Transkei region of South Africa at levels ranging from 16 to 25 μg/kg [49]. However, moniliformin detected in the present study, mean 58 μg/kg and maximum concentration 437 μg/kg, were greater than the earlier report [17]. This indicated that those samples were adversely infected by the responsible *Fusarium* species. It is revealed that moniliformin is produced by different *Fusarium* species, but mostly by *F. proliferatum* [50]. 

Among the other *Aspergillus* metabolites, 3-Nitropropionic acid had the maximum prevalence (61%), while its maximum concentration (817 μg/kg) was surpassed by viomellein (1412 μg/kg), asperfuran (2630 μg/kg), and kojic acid (4584 μg/kg) subsequently. *Penicillium* metabolites are also commonly reported as contaminants of sorghum grain [17,51]. Presently, flavoglaucin had 100% prevalence, with maximum concentration of 1870 μg/kg. However, mycophenolic acid was found to be the second most prevalent (60%), with a maximum average mean (242 μg/kg) and concentration level of 8300 μg/kg, which was the maximum levels of all metabolites detected in the current study. This metabolite was also detected in sorghum before from Ethiopia, with a maximum concentration of 198 μg/kg [17]. 

*Alternaria* species have the ability to produce different metabolites, which play an important role in fungal pathogenicity and food safety [52]. Some *Alternaria* toxins are hazardous to animal health through cytotoxicity, genotoxicity, fetotoxicity, and teratogenicity [52]. Tenuazonic acid was considered as the abundant mycotoxin in the study conducted in South Africa, contaminating sorghum samples at 100%, with the maximum levels of 293 μg/kg [35]. In accordance with that, tenuazonic acid contaminated 93% of samples in this study with an average mean of 155 μg/kg and maximum level of 1515 μg/kg, which was five times the maximum concentration detected in sorghum samples from South Africa. Consistently, this metabolite with a maximum concentration of 629 μg/kg was detected in sorghum in Ethiopian before [17]. Thereby, the study affirmed, among *Alternaria* metabolites, Ethiopian sorghum is highly contaminated with tenuazonic acid-producing *Alternaria* species.

On top of fungal metabolites and multi-mycotoxins detected in sorghum samples of the present study, bacterial metabolites, which are byproducts from bacteria that contaminate food crops and feeds, were investigated [53]. Of which the concentrations ranged from 1.06 to 14.1 μg/kg for monactin and puromycin, respectively, equivalent level of monactin was detected in the report of [17], while staurosporin 105 μg/kg was detected at the maximum level and was not presented in the current study. 

Mycotoxin contamination in other cereal crops such as maize was also reported from Ethiopia. An earlier study [54] reported that 88, 29.4, and 23.5% of maize samples were contaminated by aflatoxins, deoxynivalenol, and fumonisins, respectively. Furthermore, [55] reported total fumonisin levels of 25 to 4500 μg/kg in maize samples collected from 20 major maize-growing areas in the country, of which about 7% of the maize samples exceeded the maximum tolerable limit set by the EU in maize (>1000 μg/kg) [36] and surpassed the levels of fumonisin in the present study detected in sorghum grain. In addition, aflatoxin B_1_ was detected in the range of 3.9 to 382 μg/kg in maize grain samples [56], in which 7.7% of samples had aflatoxin B_1_ content higher than the maximum limit (5 μg/kg) set by EU in maize to be used as an ingredient in foodstuffs [56]. Recently, [57] detected about 23 metabolites in maize grain samples collected from south and south-west of Ethiopia and reported nivalenol (1052 μg/kg), deoxynivalenol (2158 μg/kg), zearalenone (2447 μg/kg), fumonisin B_2_ (2712 μg/kg), and fumonisin B_1_ (7069 μg/kg) were abundantly detected. The study revealed that *Fusarium* metabolites were prevalent among all the targeted toxins. Both incidences and levels of contaminations of aflatoxins, deoxynivalenol, and fumonisins in the present study were below the earlier reports for maize. Such variations might have been influenced owing to several factors such as the substrate, storage conditions, mold development, temperature, and relative humidity during sampled time.

### 2.4. Multi-Mycotoxins in Sorghum Samples across the Targeted Districts

Of these 94 metabolites detected, overall prevalence of metabolites across the sampled districts revealed that Goro Gutu had the maximum prevalence (76, 80.9%), followed by Doba (73, 77.5%), and samples from Fedis (68, 72.3%) and Miesso (64, 68.1%) were found to be relatively less contaminated.

Detected metabolites in all samples from each district with statistical comparisons of means and standard deviations were presented below (Table 3). Although aflatoxins are the most extensively studied fungal toxins due to their potency and prevalence in different crops, in our study, aflatoxins B_1_, B_2_, and G_1_ were detected only in samples obtained from Goro Gutu district, albeit with fewer concentrations. Aflatoxin G_2_ was not detected in the present study, congruent with the report of [41,42]. Moreover, ochratoxin A (7.86 ± 26.2 μg/kg) and deoxynivalenol (8.45 ± 30.7 μg/kg) subsequently had maximum mean and standard deviations in Doba district and relatively less in the remaining districts. In agreement with this, Ref. [17] revealed the maximum mean of deoxynivalenol (44.9 μg/kg) in their study, however, did not associate with the specific sampled areas. Generally, the concentration levels of major mycotoxins and derivatives showed less compared to the other analytes across the sampled areas.

*Fusarium* metabolites were detected in all samples with different levels of concentrations, of which fusaric acid had the maximum mean and standard deviation of 628 ± 649 μg/kg, followed by siccanol (506 ± 544 μg/kg) and moniliformin (149 ± 99.1 μg/kg) from Miesso district. This indicated that sorghum samples from that specific district abundantly infected with *Fusarium* species were responsible for the production of those metabolites, compared to the other districts. This can affirm that agro-ecology associated with other biophysical factors might have affected the fungal development and mycotoxin productions. Apart from those notoriously known *Aspergillus* metabolites such as aflatoxins, other analytes were also reported as contaminants of sorghum and its byproducts [51]. The maximum mean and standard deviation of kojic acid (546 ± 1301 μg/kg) was detected in samples from Goro Gutu district, while the maximum levels of asperfuran (307 ± 639 μg/kg) were detected in samples from Doba district. Among *Penicillium* metabolites, maximum mean and standard deviations of mycophenolic acid (763 ± 1801 μg/kg) were detected in Goro Gutu, followed by flavoglaucin (267 ± 471 μg/kg) from Doba district. Flavoglaucin was reported in sorghum malt from South Africa [51].

In the last decades, researchers have studied the occurrence of the major *Alternaria* metabolites such as alternariol, alternariolmethylether, and tentoxin [58,59,60]. However, Ref. [51] revealed tenuazonic acid as an emerging mycotoxin of *Alternaria* species, and it was detected in sorghum malt from South Africa. In the present study, tenuazonic acid was detected with the maximum mean and standard deviation (312 ± 321 μg/kg) from Miesso, followed by Fedis (180 ± 195 μg/kg). The earlier report [17] also reported tenuazonic acid levels as high as 1121 μg/kg in sorghum grain from Ethiopia. Thus, our finding affirmed that *Alternaria* species responsible for this metabolite are the most prevalent species associated with sorghum grain in Ethiopia and may warrant further study. 

Among bacterial metabolites, four of them were detected in Fedis and Miesso districts, while one and two were detected in samples from Goro Gutu and Doba, respectively, notwithstanding with lower levels compared to the fungal metabolites. Metabolites from the unspecified group were also detected, with the maximum mean and standard deviation of neoechinulin A (267 ± 451 μg/kg) from Fedis district, followed by asperglaucide (183 ± 420 μg/kg) from Miesso district. To the best of our knowledge, those metabolites were not reported in sorghum and any of food crops before in Ethiopia. However, Ref. [51] reported asperglaucide and other related metabolites such as cyclo (L-Pro-L-Tyr), cyclo(L-Pro-L-Val), and tryptophol, among others, as unspecified mycotoxins in sorghum malt from South Africa, which was in agreement with the present study. 

Generally, mycotoxin contamination across the sampled districts was not associated with the biophysical factors and agro-ecology of each district, owing to the incidences of contaminations and levels of concentration varied among the locations. 

Several factors affect mold development and mycotoxin contaminations in sorghum grain. Contamination with these toxins can occur at different points along the production chain since it is an accumulative process that may start in the field and increase during later stages including harvesting, drying, and storage. Colonization of sorghum by toxigenic fungi could be accompanied by the production of secondary metabolites including mycotoxins [17,19]. Different management approaches—such as avoiding drought under field conditions at maturity stage, use of resistant varieties, biological control, harvesting at optimum maturity, and proper drying and use of improved storage materials—reduce mold development and subsequent mycotoxin contamination in sorghum grain. Such strategies should be implemented by small scale farmers of sorghum growing to retard mycotoxin contamination in their products and reduce health risk impacts of mycotoxins at household levels.

## 3. Conclusions

The present study was focused on the post-harvest sorghum grain sample collection from five districts of East and West Hararghe zones of eastern Ethiopia for fungal species determination and multi-mycotoxin analysis. Thus, different fungal genera associated to sorghum grain were identified with the maximum prevalence of *A. niger* and *Fusarium* species. All samples were subjected to multi-mycotoxin analysis, and a total of 94 metabolites were detected and grouped in to eight categories. Metabolites of *Penicillium* fungi occurred copiously, followed by *Fusarium* metabolites, while bacterial metabolites were detected with less frequency. The levels of concentrations ranged from <LOD to 8300 μg/kg of mycophenolic acid. Regulated and emerging mycotoxins and bacterial toxins were noticed, of which major food safety concerning mycotoxins of *Alternaria*, *Fusarium,* and *Penicillium* metabolites were detected. The results indicated that there is a need to create awareness for the growers on the proper management system at pre- and post-harvest stages and further adaptation of improved storage materials to reduce fungal development and subsequent mycotoxin production to ensure safety for sorghum-based food consumers. Thus, the study is recommended for further health risk assessment of major mycotoxins associated with sorghum grain across the studied districts and potential sorghum growing areas of the country. Developing the sorghum varieties with resistance or tolerance for grain fungal infections and mycotoxin accumulations, as well as introducing and adopting the improved storage materials with adequate handling practices, should also be the prioritized strategies. 

## 4. Materials and Methods

### 4.1. Schematic Overview of the Study

The current study was comprised of sorghum sample collections and laboratory analysis of fungal species associated to sorghum grain and multi-mycotoxins analysis (Figure 3). 

This present study has been directed to find out more about major fungal species and multi-mycotoxins in sorghum grain in eastern Ethiopia. All chemicals used in this study were of analytical grade standard.

### 4.2. Sorghum Sample Collection

Sorghum sample collections were conducted in four districts (Doba, Fedis, Goro Gutu, and Miesso), i.e., two districts from East and West Hararghe zones (Figure 4 and Table 4). Districts were selected based on the sorghum production potential from eastern Ethiopia [61], and while there was no published data for Doba and Goro Gutu, those districts are abundantly producing sorghum. Then, four *kebeles* (a group of villages forming an administrative unit in Ethiopia) were selected per district and five samples were collected from each *kebele*, i.e., 20 samples per district and totally produced 80 samples were collected from the May to June 2021 cropping season. During sampling, households were randomly selected and the amounts of sorghum grain in the storage house were examined. The purpose of the study was explained, and then we requested to obtain the representative sample from their sorghum grain. A sampling spear was used to take a small portion of samples from different points of packs or lots and combine thoroughly to make an aggregated sample, and then divided into four parts equally. Then, one part weighing 500 g was taken and considered as household sample and placed in the sample bag. The sample numbers and dates were recorded, per *kebele* and district. Then, all of the samples were transported to Haramaya University for further analysis. Mycological analysis was conducted at Haramaya University, but multi-mycotoxin analysis was conducted at University of Natural Resources and Life Sciences (BOKU), Tulln, Vienna, Austria.

**Table 4 toxins-14-00473-t004:** Average rainfall and elevations of the selected districts.

Districts	Altitude (m.a.s.l)	Min and Maximum Annual Rainfall (mm)
Doba	1400–2500	550–800
Fedis	1200–1500	400–804
Goro Gutu	1250–2575	600–900
Miesso	1100–1400	400–750

Sources: Accessed from the Agriculture and Natural Resource office of the respective districts.

### 4.3. Analysis of Fungal Species Associated with Sorghum Grain Samples

Representatives of 60 seeds from each sample (*n* = 80) were randomly taken and surface-sterilized using sodium hypochlorite solution (5%) for 1–2 min and further rinsed in sterilized distilled water in two consecutive flasks for 1 min, to remove the solutions. Then, they were placed on cleaned filter paper to absorb the moisture, pre-plating. Then, for each sample, 10 seeds per plate (90 mm) in three replications were inoculated into pre-pared CZA (Czapek Agar) medium, incubated under 25 ± 2 °C, and evaluated in intervals of 24 hrs for 5 to 7 days. Incidences of infected seeds per sample were recorded daily for each district. Depending on the colony morphology, isolates were sub-cultured to newly prepared media of MEA (Malt Extract Agar) and YEA (Yeast Extract Agar) separately and incubated at 25 ± 2 °C for 5 to 7 days for subsequent characterization and taxonomic identifications. Then, further isolated fungi were identified on the basis of their micro- and macro-morphological characteristics of the standard taxonomic key using the laboratory manual series of food and indoor fungi [62]. Then, frequencies of fungal species and genera were recorded for each sampled district. 

### 4.4. Multi-Mycotoxin Analysis in Sorghum Grain Samples

From the total 500 g, a representative 50 g of sorghum grain was taken from each sample and ground to a fine flour using a laboratory mill of RT series, pulverized with extreme high speed (25,000~30,000 RPM, Ririhong, RRH-A500, Shanghai, China). Then, a representative 5 g of sorghum flour for each sample was placed into a falcon tube of 50 mL, with 20 mL of extraction solvents (acetonitrile/water/acetic acid; 79:20:1 *v*/*v*), and extracted on a GFL 3017 rotatory shaker (GFL, Burgwedel, Germany) at 180 rpm for 90 min. Then, 500 µL of dilution solvents (acetonitrile/water/acetic acid; 20:79:1 *v/v*) and the 500 µL raw extract (1:1, *v/v*) were pipetted into HPLC vials and 5 µL were subsequently injected. Then, all samples were subjected for multi-mycotoxins analysis, using a liquid chromatography-tandem mass spectrometric (LC-MS/MS) for the simultaneous determination of multiple fungal metabolites, following the protocol developed and published [63]. 

Two MS/MS transitions were acquired per analyte with the exception of moniliformin and 3-nitroropionic acid, which yield only one product ion. For confirmation of a positive identification, the ion ratio had to agree with the related values of the related authentic standard within 30% as stated in official guidelines [20], whereas for the retention time a more strict in-house criterion of ±0.03 min was applied. The limit of detection (LOD) and limit of quantification (LOQ) were calculated according to the EURACHEM guide of [63]. Quantification is based on external calibration using a serial dilution of a multi-component standard mix. Results were corrected for apparent recoveries obtained by spiking experiments. The accuracy of the method is verified by participation in proficiency testing organized by BIPEA (Genneviliers, France) on a routine basis. Currently, satisfactory z-scores between −2 and 2 have been obtained for >95% of the >1700 results submitted so far. In particular, all 27 results submitted for two samples of sorghum were in this range.

The allocation of the toxins to different species was done based on the AntiBase for microbial metabolites by [64] and supported in the papers from [65,66] on a similar topic and by our own investigations of the metabolite profile of pure cultures of toxigenic fungi. We are aware that this is still somewhat arbitrary as some given metabolites might be produced by different non-related species, e.g., fumonisin B_2_ (produced by a *Fusarium* species as well as *A. niger*).

### 4.5. Data Evaluations

For data evaluations, the peaks were integrated and linear, 1/x weighted, and calibration curves were constructed from the data obtained from the analysis of each sample type (spiked sample, neat solvent standard, spiked extract) using MultiQuantTM2.0.2 software (AB Sciex, Foster City, CA, USA) to evaluate the linearity of the method. Further data evaluation was carried out in Microsoft Excel 2010, as per published earlier [63]. 

## Figures and Tables

**Figure 1 toxins-14-00473-f001:**
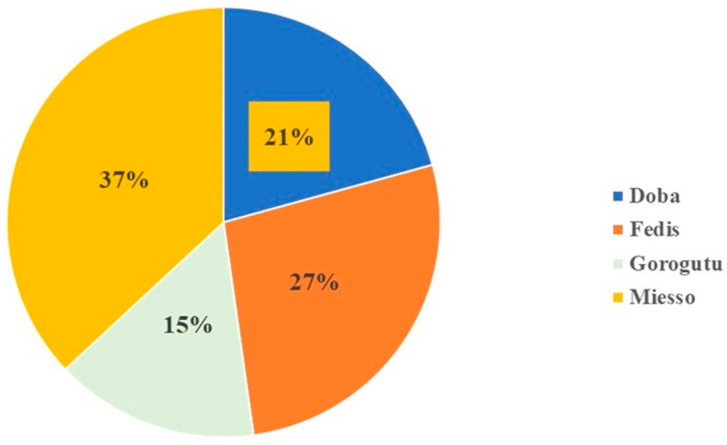
Incidences of infected samples by different fungal species across the sampled districts.

**Figure 2 toxins-14-00473-f002:**
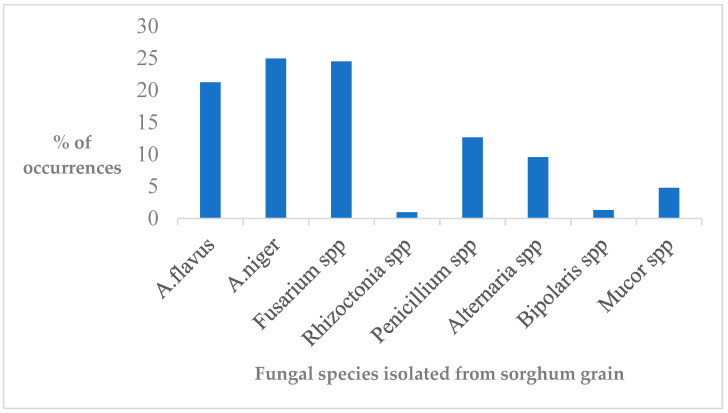
Prevalence of sorghum grain samples’ contaminated fungi.

**Figure 3 toxins-14-00473-f003:**
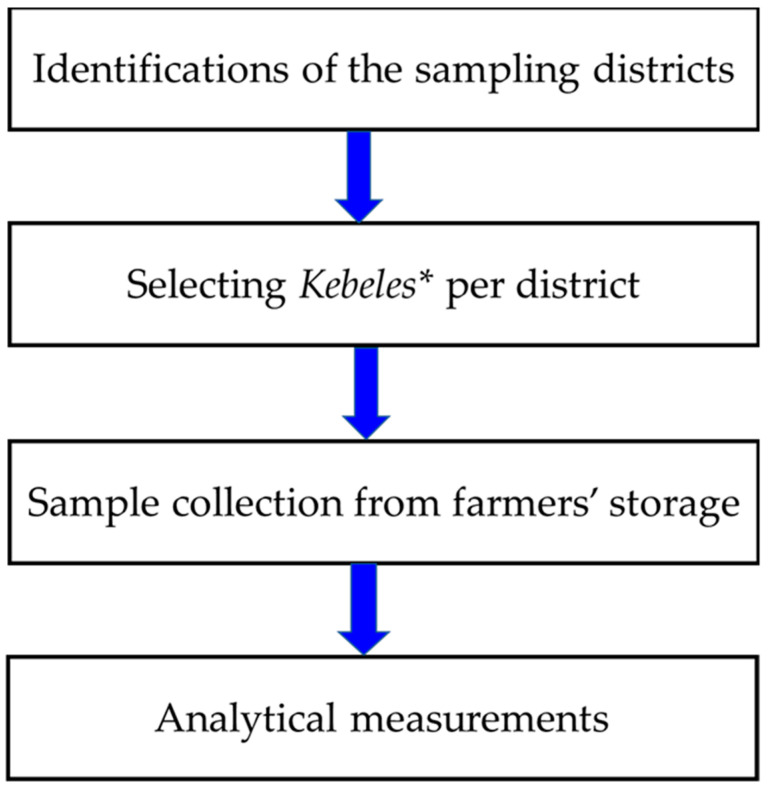
Schematic overviews of the present study. ** Kebele* is a group of villages forming an administrative unit in Ethiopia.

**Figure 4 toxins-14-00473-f004:**
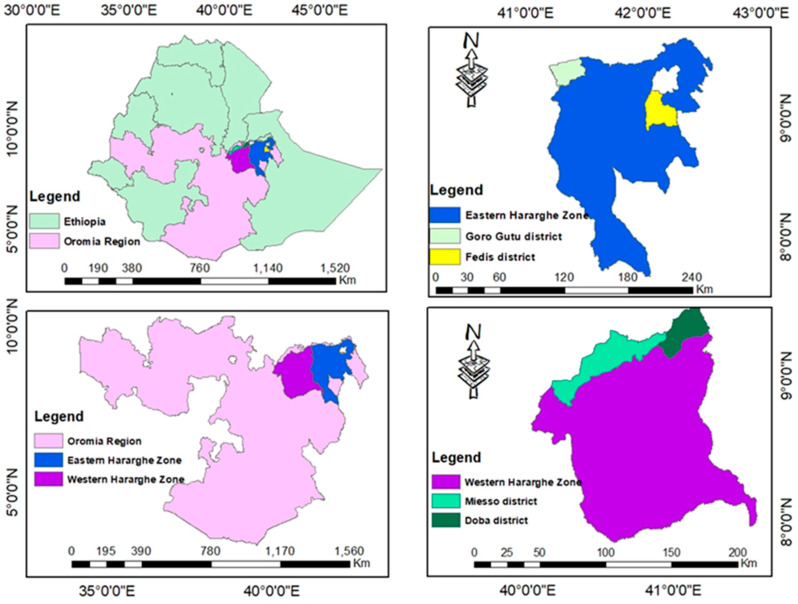
Schematic representation of selected districts for survey and post-harvest sorghum grain sample collections in East and West Hararghe zones.

**Table 1 toxins-14-00473-t001:** Incidence (%) of mycotoxins contaminated sorghum grain samples (*n* = 80) from different districts of Eastern Ethiopia, from the 2021 harvesting season.

Mycotoxins Group	Doba	Fedis	Goro Gutu	Miesso
Major mycotoxins and derivatives	40.0	15.0	40.0	65.0
*Fusarium* metabolites	95.0	100	100	100
*Aspergillus* metabolites	100	60.0	95.0	90.0
*Penicillium* metabolites	100	100	100	100
*Alternaria* metabolites	90.0	100	95.0	100
Metabolites from other fungal genera	100	100	100	100
Bacterial metabolites	60.0	55.0	80.0	95.0
Unspecified metabolites	100	100	100	100

**Table 2 toxins-14-00473-t002:** Multi-mycotoxin groups and metabolites detected in sorghum grain samples (*n* = 80) collected at post-harvest stage from eastern Ethiopia, 2021 harvesting season.

Mycotoxins Group	Specific Metabolites	% Positive Samples	Concentration (μg/kg)		
			Mean	Median	Max
Major mycotoxins and derivatives	Aflatoxin B1	1 (1.25)	0.29	<LOD	23.6
	Aflatoxin B2	1 (1.25)	0.01	<LOD	0.79
	Aflatoxin G1	1 (1.25)	0.01	<LOD	0.62
	Ochratoxin A	4 (5.0)	2.89	<LOD	115
	Ochratoxin B	4 (5.0)	0.28	<LOD	13.8
	Fumonisin B1	5 (6.25)	2.78	<LOD	112
	Fumonisin B2	2 (2.5)	0.52	<LOD	30.3
	Deoxynivalenol	5 (6.25)	2.70	<LOD	137
	Nivalenol	13 (16)	1.66	<LOD	22.3
	Monoacetoxyscirpenol	8 (10)	2.38	<LOD	59.1
	Diacetoxyscirpenol	13 (16)	0.88	<LOD	15.3
	Zearalenone	19 (24)	3.62	<LOD	121
	alpha-Zearalenol	1 (1.25)	0.08	<LOD	6.52
*Fusarium* metabolites	Moniliformin	74 (93)	58.0	25.62	437
	Beauvericin	17 (21)	0.57	<LOD	10.2
	Enniatin A1	2 (2.50)	0.02	<LOD	0.95
	Enniatin B	8 (10)	0.02	<LOD	0.65
	Enniatin B1	3 (3.75)	0.03	<LOD	1.14
	Antibiotic Y	3 (3.75)	0.15	<LOD	8.29
	Aurofusarin	15 (19)	4.79	<LOD	116
	Bikaverin	67 (84)	80.9	42.3	468
	Chrysogin	20 (25)	1.72	<LOD	34
	Deoxyfusapyron	39 (49)	14.8	<LOD	306
	Epiequisetin	4 (5.00)	0.02	<LOD	0.61
	Equisetin	23 (29)	2.42	<LOD	32.9
	Fusapyron	7 (8.80)	0.99	<LOD	27.3
	Fusaric acid	43 (54)	204	36.6	2790
	Gibepyron D	2 (2.50)	1.18	<LOD	50.9
	Sambutoxin	6 (7.50)	0.02	<LOD	0.71
	Siccanol	70 (88)	240	110	2374
	W493	6 (7.50)	1.25	<LOD	31.9
*Aspergillus* metabolites	Sterigmatocystin	24 (30)	2.80	<LOD	50.5
	Methoxysterigmatocystin	5 (6.25)	0.80	<LOD	29.7
	Versicolorin C	8 (10)	0.19	<LOD	6.08
	Averufin	23 (29)	0.66	<LOD	11.7
	8-O-Methylaverufin	12 (15)	0.32	<LOD	8.54
	Kojic acid	8 (10)	197	<LOD	4580
	3-Nitropropionic acid	49 (61)	67.8	9.07	817
	Asperflavine	27 (34)	3.83	<LOD	63
	Asperfuran	28 (35)	218	<LOD	2630
	Asperloxine A	1 (1.25)	0.04	<LOD	3.05
	Aspochracin	8 (10)	6.42	<LOD	226
	Sydonic acid	2 (2.50)	0.04	<LOD	2.77
	Viomellein	2 (2.50)	21.2	<LOD	1410
*Penicillium* metabolites	Mycophenolic acid	48 (60)	242	8.40	8300
	Mycophenolic acid IV	16 (20)	1.13	<LOD	47
	1-Deoxypebrolide	38 (48)	5.46	<LOD	127
	7-Hydroxypestalotin	6 (7.50)	0.44	<LOD	13.4
	Barceloneic acid	13 (16)	1.27	<LOD	32.8
	Chanoclavin	4 (5.00)	0.06	<LOD	1.71
	Cycloaspeptide A	5 (6.25)	1.28	<LOD	90.5
	Cyclopenin	4 (5.00)	0.08	<LOD	2.47
	Cyclopenol	5 (6.25)	0.93	<LOD	33.5
	Dechlorogriseofulvin	6 (7.50)	0.52	<LOD	23.5
	Dehydrogriseofulvin	2 (2.50)	0.01	<LOD	0.47
	F01 1358-A	1 (1.25)	0.10	<LOD	8.38
	Flavoglaucin	80 (100)	128	12.8	1870
	Griseofulvin	15 (19)	0.96	<LOD	25.3
	NP1793	5 (6.25)	3.23	<LOD	129
	O-Methylviridicatin	4 (5.00)	0.08	<LOD	5.29
	Penicillic acid	3 (3.75)	0.66	<LOD	27.9
	Quinolactacin A	4 (5.00)	0.07	<LOD	3.78
	Quinolactacin B	1 (1.25)	0.01	<LOD	0.41
	PF 1163A	2 (2.50)	0.02	<LOD	1.33
	Rugulovasine A	5 (6.25)	0.58	<LOD	24.8
*Alternaria* metabolites	Tenuazonic acid	74 (93)	155	70.5	1520
	Alternariol	47 (59)	3.39	0.73	46.4
	Alternariolmethylether	40 (50)	2.38	0.26	35.2
	Tentoxin	46 (58	1.07	0.45	9.33
	Macrosporin	57 (71)	5.61	2.32	77.1
	Radicinin	13 (16)	6.19	<LOD	126
Metabolites from other fungal genera	Abscisic acid	80 (100)	17.5	15.8	46.5
	Cytochalasin B	2 (2.50)	1.45	<LOD	109
	Destruxin A	3 (3.75)	0.06	<LOD	2.37
	Monocerin	50 (63)	6.05	0.69	119
	Preussin	5 (6.25)	0.61	<LOD	28.3
	Terphenyllin	26 (33)	61.6	<LOD	1980
	Terrein	1 (1.25)	1.49	<LOD	120
	Trichodermamide C	4 (5.00)	0.40	<LOD	11.1
Bacterial metabolites	Chloramphenicol	55 (69)	0.39	0.24	4.98
	Monactin	2 (2.50)	0.02	<LOD	1.06
	Nonactin	4 (5.00)	0.12	<LOD	5.4
	Puromycin	3 (3.75)	0.35	<LOD	14.1
Unspecific metabolites	Asperglaucide	32 (40)	76.52	<LOD	1583
	Brevianamid F	26 (33)	2.86	<LOD	110
	cyclo(L-Pro-L-Tyr)	80 (100)	33.58	20.90	574
	cyclo(L-Pro-L-Val)	8 (10)	5.71	<LOD	410
	Emodin	30 (38)	1.13	<LOD	24.6
	Fallacinol	22 (28)	3.32	<LOD	46
	N-Benzoyl-Phenylalanine	25 (31)	2.56	<LOD	51
	Neoechinulin A	44 (55)	172	7.85	2000
	Neoechinulin D	54 (68)	27.5	0.94	329
	Physcion	18 (23)	21.2	<LOD	309
	Tryptophol	11 (14)	5.08	<LOD	210

LOD = Limit of Detection.

**Table 3 toxins-14-00473-t003:** Concentrations (μg/kg) of means and standard deviations (SD) of multi-mycotoxins detected in sorghum samples of 2021, across the sampled districts.

Mycotoxin Groups	Specific Metabolites	Doba	Fedis	Goro Gutu	Miesso
		Means ± SD	Means ± SD	Means ± SD	Means ± SD
Major mycotoxins and derivatives	Aflatoxin B1	<LOD	<LOD	1.18 ± 5.28	<LOD
	Aflatoxin B2	<LOD	<LOD	0.04 ± 0.18	<LOD
	Aflatoxin G1	<LOD	<LOD	0.03 ± 0.14	<LOD
	Ochratoxin A	7.86 ± 26.2	<LOD	3.71 ± 16.57	<LOD
	Ochratoxin B	0.43 ± 1.10	<LOD	0.69 ± 3.08	<LOD
	Fumonisin B1	7.00 ± 25.45	<LOD	<LOD	4.12 ± 9.50
	Fumonisin B2	1.52 ± 6.78	<LOD	<LOD	0.58 ± 2.59
	Deoxynivalenol	8.45 ± 30.74	<LOD	<LOD	2.36 ± 7.96
	Nivalenol	<LOD	0.51 ± 2.28	1.08 ± 2.77	5.05 ± 7.13
	Monoacetoxyscirpenol	1.08 ± 4.83	2.96 ± 13.22	<LOD	5.49 ± 10.1
	Diacetoxyscirpenol	0.70 ± 3.13	0.76 ± 3.42	0.24 ± 0.92	1.84 ± 3.47
	Zearalenone	2.15 ± 7.65	0.08 ± 0.38	9.47 ± 27.97	2.77 ± 5.98
	alpha-Zearalenol	<LOD	<LOD	0.33 ± 1.46	<LOD
*Fusarium* metabolites	Moniliformin	26.5 ± 38.1	27.1 ± 22.3	29.5 ± 27.9	149 ±99.1
	Beauvericin	0.14 ± 047	0.06 ± 0.15	0.56 ± 2.19	1.53 ± 2.95
	Enniatin A1	<LOD	<LOD	0.09 ± 0.28	<LOD
	Enniatin B	0.02 ± 0.01	0.01 ± 0.02	0.07 ± 0.17	0.01 ± 0.03
	Enniatin B1	<LOD	0.02 ± 0.07	0.09 ± 0.31	<LOD
	Antibiotic Y	<LOD	0.15 ± 0.65	<LOD	0.47 ± 1.86
	Aurofusarin	1.37 ± 5.54	2.76 ± 12.36	5.13 ± 16.8	9.92 ± 26.14
	Bikaverin	23.1 ± 51.1	68.3 ± 41.8	50.2 ± 98.6	182 ± 111
	Chrysogin	2.53 ± 7.73	<LOD	2.48 ± 3.84	1.85 ± 2.62
	Deoxyfusapyron	3.53 ± 13.9	10.5 ± 20.5	4.66 ± 11.5	40.4 ± 66.1
	Epiequisetin	0.02 ± 0.08	0.02 ± 0.07	<LOD	0.05 ± 0.15
	Equisetin	0.97 ± 3.11	0.97 ± 2.26	2.92 ± 6.63	4.79 ± 8.53
	Fusapyron	<LOD	<LOD	0.50 ± 2.26	3.49 ± 6.83
	Fusaric acid	41.8 ± 115	65.4 ± 80.8	82.4 ± 184	628 ± 649
	Gibepyron D	<LOD	<LOD	<LOD	4.73 ± 14.6
	Sambutoxin	0.01 ± 0.02	<LOD	<LOD	0.06 ± 0.17
	Siccanol	46.5 ± 47.9	274 ± 502	135 ± 243	506 ± 544
	W493	0.76 ± 3.39	1.55 ± 6.93	2.53 ± 7.49	0.15 ± 0.67
*Aspergillus* metabolites	Sterigmatocystin	2.86 ± 11.27	2.18 ± 4.74	1.66 ± 4.88	4.49 ± 11.0
	Methoxysterigmatocystin	1.78 ± 6.70	0.06 ± 0.27	1.38 ± 4.33	<LOD
	Versicolorin C	0.34 ± 1.36	0.11 ± 0.37	0.29 ± 0.72	0.04 ± 0.19
	Averufin	0.73 ± 2.64	0.48 ± 0.82	0.82 ± 2.29	0.59 ± 1.44
	8-O-Methylaverufin	0.13 ± 0.52	0.07 ± 0.22	0.37 ± 1.51	0.73 ± 1.98
	Kojic acid	235 ± 1024	<LOD	546 ± 1301	9.31 ± 34.0
	3-Nitropropionic acid	85.0 ± 115	9.47 ± 40.36	84.2 ± 124	92.7 ± 199
	Asperflavine	4.41 ± 10.23	9.55 ± 18.35	1.01 ± 1.66	0.34 ± 1.00
	Asperfuran	307 ± 639	92.2 ±149	235 ± 419	238 ± 466
	Asperloxine A	0.15 ± 0.68	<LOD	<LOD	<LOD
	Aspochracin	1.77 ± 5.54	11.3 ± 50.6	4.12 ± 15.1	8.49 ± 36.2
	Sydonic acid	<LOD	0.14 ± 0.62	0.03 ± 0.11	<LOD
	Viomellein	70.6 ± 316	<LOD	14.1 ± 63.1	<LOD
*Penicillium* metabolites	Mycophenolic acid	167 ± 273	33.8 ±138	763 ±1801	3.07 ± 5.57
	Mycophenolic acid IV	0.70 ± 1.37	0.32 ± 1.45	3.51 ± 10.4	<LOD
	1-Deoxypebrolide	8.50 ± 8.53	0.12 ± 0.53	13.21 ± 27.6	<LOD
	7-Hydroxypestalotin	<LOD	<LOD	1.75 ± 3.34	<LOD
	Barceloneic acid	<LOD	0.75 ± 2.37	<LOD	4.31 ± 7.34
	Chanoclavin	0.10 ± 0.31	<LOD	0.05 ± 0.24	0.08 ± 0.38
	Cycloaspeptide A	<LOD	4.55 ± 20.2	<LOD	0.60 ± 2.23
	Cyclopenin	0.03 ± 0.16	0.12 ± 0.55	0.16 ± 0.55	<LOD
	Cyclopenol	1.04 ± 4.67	0.22 ± 0.97	2.48 ± 7.73	<LOD
	Dechlorogriseofulvin	1.57 ± 5.33	0.27 ± 1.19	0.11 ± 0.48	0.14 ± 0.63
	Dehydrogriseofulvin	0.01 ± 0.03	<LOD	0.02 ± 0.11	<LOD
	F01 1358-A	<LOD	<LOD	0.42 ± 1.87	<LOD
	Flavoglaucin	267 ± 471	118 ± 181	91.3 ± 148	34.5 ± 64.5
	Griseofulvin	2.51 ± 6.68	0.52 ± 1.00	0.57 ± 1.13	0.23 ± 1.02
	NP1793	0.92 ± 2.86	6.45 ± 28.9	1.85 ± 8.29	3.71 ± 16.6
	O-Methylviridicatin	<LOD	0.33 ± 1.19	0.01 ± 0.04	<LOD
	Penicillic acid	1.56 ± 6.24	<LOD	1.09 ± 4.90	<LOD
	Quinolactacin A	0.21 ± 0.84	<LOD	0.05 ± 0.18	<LOD
	Quinolactacin B	0.02 ± 0.09	<LOD	<LOD	<LOD
	PF 1163A	<LOD	<LOD	0.09 ± 0.03	<LOD
	Rugulovasine A	1.41 ± 5.55	0.76 ± 2.35	0.17 ± 0.77	<LOD
*Alternaria* metabolites	Tenuazonic acid	47.9 ± 43.5	180 ± 195	79.4 ± 96.1	312 ± 321
	Alternariol	0.58 ± 1.09	3.45 ± 7.22	3.55 ± 10.32	5.99 ± 10.1
	Alternariolmethylether	0.39 ± 1.04	2.64 ± 7.76	2.72 ± 7.30	3.78 ± 6.82
	Tentoxin	0.17 ± 0.30	1.46 ± 1.57	0.22 ± 0.64	2.45 ± 2.51
	Macrosporin	0.88 ± 1.89	11.0 ± 17.1	0.75 ± 0.94	9.78 ± 5.73
	Radicinin	0.46 ± 2.05	12.2 ± 29.4	9.34 ± 23.13	2.82 ± 10.23
Metabolites from other fungal genera	Abscisic acid	16.6 ± 6.66	19.4 ± 8.42	14.5 ± 4.58	19.49 ± 4.56
	Cytochalasin B	<LOD	0.36 ± 1.61	5.43 ± 24.3	<LOD
	Destruxin A	0.19 ± 0.61	0.08 ± 0.35	<LOD	<LOD
	Monocerin	8.68 ± 27.2	7.16 ± 7.92	4.68 ± 9.50	3.68 ± 7.17
	Preussin	0.63 ± 1.95	1.42 ± 6.33	0.25 ± 1.13	0.16 ± 0.69
	Terphenyllin	13.2 ± 25.4	136 ± 438	10.9 ±21.9	86.3 ± 210
	Terrein	5.99 ± 26.8	<LOD	<LOD	<LOD
	Trichodermamide C	0.56 ± 1.75	0.55 ± 2.47	0.49 ± 2.23	<LOD
Bacterial metabolites	Chloramphenicol	0.15 ± 0.19	0.12 ± 0.16	0.32 ± 0.31	1.00 ± 1.09
	Monactin	<LOD	0.03 ± 0.14	<LOD	0.05 ± 0.23
	Nonactin	<LOD	0.27 ± 1.21	<LOD	0.20 ± 0.76
	Puromycin	0.32 ± 1.41	0.70 ± 3.14	<LOD	0.39 ± 1.75
Unspecific metabolites	Asperglaucide	58.0 ± 135	63.1 ±130	1.74 ± 3.86	183 ± 420
	Brevianamid F	1.01 ± 2.14	6.22 ± 24.4	1.51 ± 2.18	2.71 ± 2.79
	cyclo(L-Pro-L-Tyr)	19.9 ± 12.4	48.9 ± 124	22.4 ± 12.3	43.1 ±27.8
	cyclo(L-Pro-L-Val)	<LOD	20.5 ± 91.6	<LOD	2.34 ± 3.31
	Emodin	0.62 ± 1.33	2.99 ± 5.81	0.67 ± 1.39	0.26 ± 0.46
	Fallacinol	3.18 ± 7.23	6.98 ± 12.6	2.64 ± 8.62	0.51 ± 1.66
	N-Benzoyl-Phenylalanine	3.39 ± 5.64	0.32 ± 0.60	4.70 ± 11.8	1.84 ± 4.08
	Neoechinulin A	250 ± 480	267 ± 451	109 ± 176	59.9 ± 136
	Neoechinulin D	33.4 ± 76.2	49.6 ± 93.4	16.9 ± 27.2	9.99 ± 21.2
	Physcion	28.7 ± 69.4	47.0 ± 78.0	6.37 ± 19.6	2.85 ± 12.8
	Tryptophol	5.15 ± 10.2	11.5 ± 47.0	3.73 ± 7.78	<LOD

## Data Availability

We didn’t report any data before.

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
