# Peer review of "Fungal Species and Multi-Mycotoxin Associated with Post-Harvest Sorghum (Sorghum bicolor (L.) Moench) Grain in Eastern Ethiopia"

_toxins, 2022, doi:10.3390/toxins14070473_

Round 1

Reviewer 1 Report

The article "Fungal Species and Multi-mycotoxin Associated with Post-harvest Sorghum (Sorghum bicolor (L.) Moench) Grain in

Eastern Ethiopia" is interesting and very important for agriculture.

However, I have a few remarks that prevent its publication in its current form.

The main remark is a very brief and uninformative description of the materials and research methods. It is not clear how fungal species were identified, how the belonging of toxins to specific fungal species was determined.

Other notes:

Missing authors and affiliations

In my opinion, "Fungal species" is not a very good wording for the keywords section.

Line 70 Fusarium in italics, remove ";" after "as"

Line 101 - species and generic names are mixed. Write in full Aspergillud flavus, A. niger.

Figure 2 should be in Mukor sp? It is necessary to sign the axes in figure 2

Line 215, 222, 242 italicized Latin names

Conclusion needs to be made clearer and more concise, reflecting exactly the results obtained by the authors. The first few phrases are not conclusions from this work, they can be removed.

Author Response

Author's Reply to the Review Report (Reviewer 1)

Comments and Suggestions for Authors

The article "Fungal Species and Multi-mycotoxin Associated with Post-harvest Sorghum (Sorghum bicolor (L.) Moench) Grain in Eastern Ethiopia" is interesting and very important for agriculture.

However, I have a few remarks that prevent its publication in its current form.

The main remark is a very brief and uninformative description of the materials and research methods. It is not clear how fungal species were identified, how the belonging of toxins to specific fungal species was determined.

Responses:

  • Fungal species identifications as follows:

Representative of 60 seeds from each sample (N= 80) were randomly taken and surface sterilized using sodium hypochlorite solution (5%) for 1-2 min and further rinsed in sterilized distilled water in two consecutive flasks for 1 min, to remove the solutions. Then placed on cleaned filter paper to absorb the moisture, pre-plating. Then for each sample, 10 seeds per plate (90 mm) in three replications were inoculated into pre-pared CZA (Czapek Agar) medium and incubated under 25±2 °C and evaluated in the intervals of 24 hrs for 5 to 7 days. Incidences of infected seed per sample was recorded daily for each district. Depends on the colony morphology, isolates were sub-cultured to newly prepared media of MEA (Malt Extract Agar) and YEA (Yeast Extract Agar) separately and incubated at 25±2 °C for 5 to 7 days for subsequent characterization and taxonomic identifications. Then further isolated fungi were identified on the basis of their micro and macro-morphological characteristics of the standard taxonomic key using the laboratory manual series of food and indoor fungi [63]. Then frequencies of fungal species and genera were recorded for each sampled district.

  • Belongness of toxins to specific fungal species group:

The allocation of the toxins to different species was done based on the AntiBase for microbial metabolites by [65] and supported in the papers from [66,67] on a similar topic and by our own investigations of the metabolite profile of pure cultures of toxigenic fungi. We are aware that, this is still somewhat arbitrarily as some given metabolites might be produced by different non-related species e.g., fumonisin B2 (produced by a Fusarium species as well as A niger).

  • Notes: Those references are, at their respective numbers and sections in the main texts.

Other notes:

Missing authors and affiliations

In my opinion, "Fungal species" is not a very good wording for the keywords section.

Responses: Fungal species, deleted from the Keywords.

Line 70 Fusarium in italics, remove ";" after "as"

Responses: Fusarium, italicized and “;” after as, removed.

Line 101 - species and generic names are mixed. Write in full Aspergillus flavus, A. niger.

Responses: Aspergillus flavus, A. niger.

Figure 2 should be in Mukor sp? It is necessary to sign the axes in figure 2

Response: Mucor spp, edited. Axis added, in Figure 2.

Line 215, 222, 242 italicized Latin names

Responses: All pathogen names, are italicized.

Conclusion needs to be made clearer and more concise, reflecting exactly the results obtained by the authors. The first few phrases are not conclusions from this work, they can be removed.

Responses: Few sentences at the beginning of the conclusions were deleted and the conclusion modified as follows in the document:

The present study was focused on the post-harvest sorghum grain samples collection from five districts of East and West Hararghe zones of eastern Ethiopia for fungal species determination and multi-mycotoxin analysis. Thus, different fungal genera associated to sorghum grain were identified with the maximum prevalence of A. niger and Fusarium specie. All samples were subjected for multi-mycotoxin analysis and a total of 94 metabolites were detected and grouped in to 8 categories. Metabolites of Penicillium fungi were copiously occurred, followed by Fusarium metabolites, while bacterial metabolites were detected in less frequency. The levels of concentrations were ranged from <LOD to 8300 μg/kg of mycophenolic acid. Regulated and emerging mycotoxins, and bacterial toxins were noticed. Of which major food safety concerning mycotoxins of Alternaria, Fusarium and Penicillium metabolites were detected. The results indicated that, there is a need to create awareness for the growers, on the proper management system at pre-and post-harvest stages and further adaptation of improved storage materials to reduce fungal development and subsequent mycotoxin production to ensure safety for sorghum-based food consumers. So, the study is recommended for further health risk assessment of major mycotoxin associated to sorghum grain across the studied districts and potential sorghum growing areas of the country. Developing the sorghum varieties with resistant or tolerant for grain fungal infections and mycotoxin accumulations, as well as introducing and adopting the improved storage materials with adequate handling practices should also the priority strategies.

Reviewer 2 Report

This is a promising work, which determined the major fungal species and multi-mycotoxins associated with post-harvest sorghum grain in eastern Ethiopia, covering East and West Hararghe Zones. It can be improved through the following:
a) Introduction is very good, however, it will be useful for authors, prior to merge paragraphs 1 and 2, to become one paragraph. Then, new paragraph 2 should in 4-5 sentences identify the various post-harvest challenges that confront sorghum product, globally, then narrowed down to Ethiopia, where the last sentence should bring in fungal concerns, e.g., mycotoxins as examples. Then, paragraph 3 should then begin with Line 51, which should merge with paragraph starting line 62.
Please, paragraph starting line 78, should merge with paragraph starting line 90, because both contains the case of rationale for this study, prior to the objective statement. So in all, introduction should have 4 distinct paragraphs. Reviewer will check this in the revised manuscript.

2. Results and discussion very comprehensive and interactive. It is essential for table 3 to have the superscript indicating significant differences where specific mycotoxin metabolites were detected in two or more sampled districts. Please, reflect this in the statistical analysis section that one-way ANOVA was used to determine differences in specific mycotoxin metabolites across sampled districts. Please, check the SDs, it is somewhat odd that SDs will be higher than the mean values?? Is it correct?
 3. Conclusions is ok. Clearly indicate where the direction of future studies is , it is mentioned, but clearly indicate it, 'Given the findings of this current work, it will be useful for future studies to be directed towards ....

4. Materials and methods, is very ok. However, it will help if the authors can begin this section with a new subsection captioned, 'Schematic overview of the experimental program', which will comprise 3 sentences, and supported by a flow diagram, which should show:
Identification of sampling locations> Defining the kebele and adequacy of samples> Collection of samples from farmers' storage> Analytical measurements >>> (Then spread the tentacles based on the analyses performed). So sentence one should introduce the schematic diagram, Figure x shows the schematic..., which depicts the major stages of this work, from ....to analytical measurements. Sentence two, For emphasis, this work has been directed to find out more about...... Sentence three, All
chemicals used at this study were of analytical grade standard.
Reviewer looks forward to seeing this, it is necessary to guide the readers of this work.
In 4.1, which will now be 4.2, a) explain how the sampling locations were identified, b) define what what kebele is, with reference, c) Please explain the process of sample collection, (Please provide adequate details, tell us what you did, present it as succinctly, yet thoroughly as possible).

The reviewer believes with these, there would be an improved and more robust revised manuscript. Looking forward to the revised manuscripot.
Very best regards :)

Author Response

Author's Reply to the Review Report (Reviewer 2)

Comments and Suggestions for Authors

This is a promising work, which determined the major fungal species and multi-mycotoxins associated with post-harvest sorghum grain in eastern Ethiopia, covering East and West Hararghe Zones.

It can be improved through the following:
a) Introduction is very good, however, it will be useful for authors, prior to merge paragraphs 1 and 2, to become one paragraph. Then, new paragraph 2 should in 4-5 sentences identify the various post-harvest challenges that confront sorghum product, globally, then narrowed down to Ethiopia, where the last sentence should bring in fungal concerns, e.g., mycotoxins as examples. Then, paragraph 3 should then begin with Line 51, which should merge with paragraph starting line 62.
Please, paragraph starting line 78, should merge with paragraph starting line 90, because both contains the case of rationale for this study, prior to the objective statement. So in all, introduction should have 4 distinct paragraphs. Reviewer will check this in the revised manuscript.

Responses: Paragraphs 1 and 2 merged.

Paragraph 2, inserted.

Paragraphs, started with Line 51 merged with a paragraph starting with line 62, produce paragraph 3.

Paragraph, stared with Line 78 merged with paragraph stated with line 90, and produces paragraph 4.

Then, introduction have four separate paragraphs in the revised version.

  1. Results and discussion very comprehensive and interactive. It is essential for table 3 to have the superscript indicating significant differences where specific mycotoxin metabolites were detected in two or more sampled districts. Please, reflect this in the statistical analysis section that one-way ANOVA was used to determine differences in specific mycotoxin metabolites across sampled districts. Please, check the SDs, it is somewhat odd that SDs will be higher than the mean values?? Is it correct?

Responses: Yes, because in some samples there was small concentration detected, in some samples not detected, and that leads to an inflated SDs.

When there was uniform concentration in several samples, the SDs were either equivalent or less than the average mean values, per individual district.

e.g. Moniliformin, for Fedis, Goro Gutu and Miesso.

  1. Conclusions is ok. Clearly indicate where the direction of future studies is, it is mentioned, but clearly indicate it, 'Given the findings of this current work, it will be useful for future studies to be directed towards ....

Responses: Conclusion was modified as follows:

The present study was focused on the post-harvest sorghum grain samples collection from five districts of East and West Hararghe zones of eastern Ethiopia for fungal species determination and multi-mycotoxin analysis. Thus, different fungal genera associated to sorghum grain were identified with the maximum prevalence of A. niger and Fusarium specie. All samples were subjected for multi-mycotoxin analysis and a total of 94 metabolites were detected and grouped in to 8 categories. Metabolites of Penicillium fungi were copiously occurred, followed by Fusarium metabolites, while bacterial metabolites were detected in less frequency. The levels of concentrations were ranged from <LOD to 8300 μg/kg of mycophenolic acid. Regulated and emerging mycotoxins, and bacterial toxins were noticed. Of which major food safety concerning mycotoxins of Alternaria, Fusarium and Penicillium metabolites were detected. The results indicated that, there is a need to create awareness for the growers, on the proper management system at pre-and post-harvest stages and further adaptation of improved storage materials to reduce fungal development and subsequent mycotoxin production to ensure safety for sorghum-based food consumers. So, the study is recommended for further health risk assessment of major mycotoxin associated to sorghum grain across the studied districts and potential sorghum growing areas of the country. Developing the sorghum varieties with resistant or tolerant for grain fungal infections and mycotoxin accumulations, as well as introducing and adopting the improved storage materials with adequate handling practices should also the priority strategies.

  1. Materials and methods, is very ok. However, it will help if the authors can begin this section with a new subsection captioned, 'Schematic overview of the experimental program', which will comprise 3 sentences, and supported by a flow diagram, which should show:
    Identification of sampling locations> Defining the kebele and adequacy of samples> Collection of samples from farmers' storage> Analytical measurements >>> (Then spread the tentacles based on the analyses performed). So sentence one should introduce the schematic diagram, Figure x shows the schematic..., which depicts the major stages of this work, from. sampling...to analytical measurements. Sentence two, For emphasis, this work has been directed to find out more about...... Sentence three, All
    chemicals used at this study were of analytical grade standard.

Responses: for this comments presented as follows:

4.1. Schematic overview of the study

The current study was comprised of sample collections and laboratory analysis of fungal species associated to sorghum grain and multi-mycotoxins (Figure 3). 

Figure 3. Schematic overviews of the present study. *Kebele is a group of villages forming an administrative unit in Ethiopia.

This present study has been directed to find out more about major fungal species in sorghum grain in eastern Ethiopia and multi-mycotoxins. All chemicals used at this study were of analytical grade standard.

Reviewer looks forward to seeing this, it is necessary to guide the readers of this work.
In 4.1, which will now be 4.2, a) explain how the sampling locations were identified, b) define what kebele is, with reference, c) Please explain the process of sample collection, (Please provide adequate details, tell us what you did, present it as succinctly, yet thoroughly as possible).

The reviewer believes with these, there would be an improved and more robust revised manuscript. Looking forward to the revised manuscript.
Very best regards :)

Response:

A). Some districts were selected based on the sorghum production potential.

In fact, it was not supported by references, both Goro Gutu and Doba, districts are also known by sorghum production and very potential districts.

  1. b) *Kebele is a group of villages forming an administrative unit in Ethiopia.

  1. c) The process of sample collections:

Sorghum sample collections were conducted in four districts (Doba, Fedis, Goro Gutu, and Miesso), i.e., two districts from East and West Hararghe zones (Figure 4 & Table 4). Districts were selected based on the sorghum production potential from eastern Ethiopia [62], while there was no published data for Doba and Goro Gutu, those districts are abundantly producing sorghum. Then, four kebeles (a group of villages forming an administrative unit in Ethiopia) were selected per district and five samples were collected from each kebele, i.e., 20 samples per district and totally produced 80 samples were collected from May to June of 2021 cropping season. During sampling, households were randomly selected and the amounts of sorghum grain in the storage house were examined, and the purpose of the study was explained and then requested to obtained the representative sample from their sorghum grain. Sampling spear was used to take small portion of samples from different points of packs or lots and combine, thoroughly to make an aggregated samples and then divided into four parts equally. Then one part, weighted 500 grams was taken and considered as household sample, placed in the sample bag, recorded the sample numbers and date, per kebele and district. Then the whole samples transported to Haramaya University for further analysis. Mycological analysis conducted at Haramaya University, however multi-mycotoxin analysis was conducted at University of Natural Resources and Life Sciences (BOKU), Tulln, Vienna, Austria.

Reviewer 3 Report

Thank you for the opportunity to review this article.

The authors evaluated the: The manuscript deals with the topic: Fungal Species and Multi-mycotoxin Associated with Post-harvest Sorghum (Sorghum bicolor (L.) Moench) Grain in Eastern Ethiopia

The manuscript deals with interesting topic, but in my opinion the manuscript requires some improvement before publication.

Specific remarks:

Line 58-59. You can add from the recent review article by Agriopoulou et al (2020) with title Advances in Occurrence, importance and mycotoxin control strategies: Prevention and detoxification in foods (Agriopoulou, S.; Stamatelopoulou, E.; Varzakas, T. Advances in Occurrence, Importance, and Mycotoxin Control Strategies: Prevention and Detoxification in Foods. Foods 2020, 9, 137)

Line 82.SSA countries...What is SSA?

Line 201.AFG2...Line 63 AfB1...Please write in the same form

Line 215.Fusarium metabolites...Please write all the pathogens in italics in the whole text. The same for Aspergillus metabolites, Alternaria metabolites  in the whole text

Line 223.enniatins (ENNs)...Only one time we use this.Since we have once written the abbreviation together with the whole word we do not rewrite it this way. We write only the abbreviation. Please apply it throughout the text

Line 231.You can also add from the recent review article by Agriopoulou et al (2016) with title Enniatins: An Emerging Food Safety Issue (Agriopoulou, S. Enniatins: An emerging food safety issue. EC Nutr. 2016, 3, 1142–1146.)

Line 241.F. proliferatum...Please in italics

For all references need to write the abbreviation of the journal 

Reference 14 and  49. Please write with correct form

Author Response

Author's Reply to the Review Report (Reviewer 3)

Comments and Suggestions for Authors

Thank you for the opportunity to review this article.

The authors evaluated the: The manuscript deals with the topic: Fungal Species and Multi-mycotoxin Associated with Post-harvest Sorghum (Sorghum bicolor (L.) Moench) Grain in Eastern Ethiopia

The manuscript deals with interesting topic, but in my opinion the manuscript requires some improvement before publication.

Specific remarks:

Line 58-59. You can add from the recent review article by Agriopoulou et al (2020) with title Advances in Occurrence, importance and mycotoxin control strategies: Prevention and detoxification in foods (Agriopoulou, S.; Stamatelopoulou, E.; Varzakas, T. Advances in Occurrence, Importance, and Mycotoxin Control Strategies: Prevention and Detoxification in Foods. Foods 2020, 9, 137):

Responses: Line 58-59. Fungal species orders, from Aspergillus, Penicillium and Fusarium species, re-arranged to… Aspergillus, Fusarium, and penicillium species.

The following points inserted from line 59 and extended the ## of lines, from the earlier version.

Mycotoxin effects in humans and animals following direct exposures varies in terms of their toxicity, e.g., carcinogenic, endocrine disorders, teratogenic, mutagenic, hemorrhagic, estrogenic, hepatotoxic, nephrotoxic, and immunosuppressive [16].

Line 82.SSA countries...What is SSA?

Responses: Line 82. Modified as; In some sub-Saharan African countries (SSA),…..

Line 201.AFG2...Line 63 AfB1...Please write in the same form

Responses: Line 63, modified as…. AFB1, and Line 201, as its AFG2.

Line 215.Fusarium metabolites...Please write all the pathogens in italics in the whole text. The same for Aspergillus metabolites, Alternaria metabolites in the whole text

Responses: Line 215. … “Fusarium” metabolites…, italic. And all the name of the pathogens, Aspergillus, Alternaria, Fusarium and Penicillium species, which were not italic, becomes italic in the current version and blue colored.

Line 223.enniatins (ENNs)...Only one time we use this. Since we have once written the abbreviation together with the whole word we do not rewrite it this way. We write only the abbreviation. Please apply it throughout the text

Responses: From line 223-226, I have deleted all of the abbreviations for those compounds, for the consistence’s. Then, the original lines may re shuffled.

Line 231.You can also add from the recent review article by Agriopoulou et al (2016) with title Enniatins: An Emerging Food Safety Issue (Agriopoulou, S. Enniatins: An emerging food safety issue. EC Nutr. 2016, 3, 1142–1146.)

Responses: yes, its added

Previously [48], revealed the likely frequent contamination of enniatins in cereals and cereals-based foods, and urged further investigation of such mycotoxins across diverse geographical regions with different climatic conditions from all over the world. Thus, the present study affirmed the contamination of enniatins as an emerging mycotoxin in sorghum grain from Ethiopia.

Line 241.F. proliferatum...Please in italics

Responses: F. proliferatum, italic.

For all references need to write the abbreviation of the journal 

Responses: All journals, abbreviated in the reference sections.

Reference 14 and 49. Please write with correct form

Responses: References #14, and 49, corrected and moved down, #14 become 17, and #49 become 59, due to some other references being added in the new version beyond them.

Reviewer 4 Report

How were the fungi identified? This should be described in methods.

What are the regulatory limits and how does this relate with the current findings?

Could you highlight the production methods that could reduce major mycotoxins in sorghum?

How does the contamination compare with other crops in Ethiopia?

Author Response

Author's Reply to the Review Report (Reviewer 4)

Comments and Suggestions for Authors

How were the fungi identified? This should be described in methods.

Responses: Fungi were identified using the laboratory manual series [63].

What are the regulatory limits and how does this relate with the current findings?

 Responses: Regulatory limits were compared with the Aflatoxin B1, DON, OTA, Fumonisin B1 and B2 in the document, being as a major concerned mycotoxin.

Could you highlight the production methods that could reduce major mycotoxins in sorghum?

Responses: the following paragraph incorporated in the document.

Several factors affect mold development and mycotoxin contaminations in sorghum grain. Contamination with these toxins can occur at different points along the production chain since it is an accumulative process that may start in the field and increase during later stages including harvesting, drying, and storage. Colonization of sorghum by toxigenic fungi could be accompanied by the production of secondary metabolites including mycotoxins [17,19]. So that different management approaches like avoiding drought under field conditions, use of resistant varieties, biological control, harvesting at optimum maturity and proper drying and use of improved storage materials reduce mold development and subsequent mycotoxin contamination in sorghum grain. Such strategies should be implemented by small scale farmers of sorghum growing to retard mycotoxin contamination in their products and reducing health risk impacts of mycotoxin at household level.

How does the contamination compare with other crops in Ethiopia?

Responses: Mycotoxin contamination of sorghum in the current finding is compared with the previous reports of mycotoxin in maize in Ethiopia. Being maize and sorghum are the main food crops in the major part of Ethiopia and the targeted studied areas too.

  • Then, the following paragraph incorporated in the text.

Mycotoxin contamination in other cereal crops like maize also reported from Ethiopia. Of which the earlier study [55] reported 88, 29.4 and 23.5 % of maize samples were contaminated by aflatoxins, deoxynivalenol and fumonisins, respectively.  Furthermore [56], reported total fumonisins levels of 25 to 4500 μg/kg in maize samples collected from 20 major maize growing areas in the country, of which about 7% of the maize samples exceeded the maximum tolerable limit set by the EU in maize (> 1000 μg/kg [37] and surpassed the levels of fumonisin in the present study detected in sorghum grain.  In addition, AFB1 was detected in the range of 3.9–381.6 μg/kg in maize grain samples [57], in which 7.7% of samples had AFB1 content higher than the maximum limit (5 μg/kg) set by EU for maize to be use as an ingredient in foodstuffs [37]. Recently [58] used to detect about 23 metabolites in maize grain samples collected from South and South West of Ethiopia and reported nivalenol (1,052 μg/kg), deoxynivalenol (2,158 μg/kg), zearalenone (2,447 μg/kg), fumonisin B2 (2,712 μg/kg), and fumonisin B1 (7,069 μg/kg) were abundantly detected. The study revealed that, Fusarium metabolites were the prevalent among all the targeted toxins. In which both incidences and levels of contaminations of aflatoxins, deoxynivalenol and fumonisins in the present study were below the earlier reports in maize. Such variations might have been influenced owing to several factors like, the substrate, storage conditions, mold development, temperature and relative humidity during sampled time.

Round 2

Reviewer 1 Report

In my opinion the article can be published in its current form

Reviewer 2 Report

Author revised their work well.

Reviewer is very satisfied.

It is now acceptable for publication.